# OpenReview forum: "Semantic Representation Attack against Aligned Large Language Models"
_NeurIPS.cc/2025/Conference — NeurIPS 2025 poster_

### Official Review · Reviewer_uz6Q · 2025-06-30

**Clarity:** 4
**Significance:** 2
**Originality:** 3
**Rating:** 5
**Confidence:** 4

**Summary:**

This paper proposes a novel jailbreaking attack paradigm and an algorithm called Semantic Representation Heuristic Search (SRHS). This paper pointed out that current methods measure and optimize the effectiveness of an adversarial prompt by comparing it to a fixed, exact positive response, e.g., "Sure, here is ...". This paradigm suffers from limited convergence, unnatural prompts and high computation cost. The proposed method exploits the semantic representation space and generate adversarial prompts that is semantic coherent to the original malicious question. This paper also provides theoretical analysis and extensive experiments to validate their methods.

**Questions:**

In the limitation part, the authors claim that *the proposed approch requires logit access to the victim model* (which implies that SRHS cannot be applied to blalck-box LLMs). However, it is a widely acknowledged that a large propotion of the adversarial prompts can transfer across different LLMs. My first question is:

1. What is the transferability of the proposed method?

As for the "unknown distribution" issue, it is commonly believed that the models within the same family with different scale (e.g., Qwen2.5-0.5B and Qwen2.5-70B) share a similar output distribution (The effectiveness of speculative inference has validated this belief in some sense). I am very curious about the transferability of the proposed method across such models. My second question, which might seem alike to my first one, would be:

2. Could the authors please design and conduct auxiliary experiment to validate my conjecture (i.e., the proposed method would transfer better within models in a same family)?

**Ethical Concerns:**

["NO or VERY MINOR ethics concerns only"]

**Final Justification:**

I would like to thank the authors for their detailed response, which has addressed the majority of my concerns. In particular, the authors have conducted supplementary experiments to verify my conjecture regarding transferability both within and across model families. That said, several issues remain—for instance, there is a lack of experiments involving state-of-the-art closed-source models, e.g., the online version of GPT, Gemini, and DeepSeek. Nevertheless, given that these models are aligned by mature, commercialized teams, launching jailbreaking attacks against such systems may place academic researchers at an unfair disadvantage. Overall, I find the present paper to be novel (introducing a new paradigm), technically solid, and theoretically intuitive. As such, I have revised my score to 5.

**Limitations:**

Yes, the authors have adequatedly addressed the limitations in page 9. See the questions part for a possible solution for black-box utility of this method.

**Quality:**

3

**Strengths And Weaknesses:**

**Strength**
1. The representation of this paper is impressive. The formulation of the jailbreaking attack and the follow-up analysis is easy-to-understand.
2. This paper provides theoretical anlysis, where is not commonly seen in the researches of jailbreaking attacks.
3. This paper conduct comprehensive experiments across a considerable number of popular 7B-level LLMs and jailbreaking attack methods.

**Weaknesses**
1. My main and only concern is the significance of this paper. The LLMs used in the experiments has already been proven, by many previous studies, to be very fragile. I have conducted PAIR and AutoDAN on Llama2-7B and Vicuna-7B and -13B. This two methods reach >90% ASR. In fact, it is not hard to jailbreak these models even using DAN-like adversarial prompts. I suggest the authors to challenge (in a safe circumstances) the SOTA close-source LLM like GPT, Claude, and Deepseek for better evaluations.

---

> ### Author Rebuttal · Authors · 2025-07-29
>
> Dear Reviewer uz6Q,
>
> Thank you for your positive feedback on our paper's presentation, theoretical analysis, and comprehensive experiments. We appreciate your thoughtful comments and your main concern regarding the significance of our work, which we will address below.
>
> ---
>
> **W1:** My main and only concern is the significance of this paper. The LLMs used in the experiments has already been proven, by many previous studies, to be very fragile. I have conducted PAIR and AutoDAN on Llama2-7B and Vicuna-7B and -13B. This two methods reach >90% ASR. In fact, it is not hard to jailbreak these models even using DAN-like adversarial prompts. I suggest the authors to challenge (in a safe circumstances) the SOTA close-source LLM like GPT, Claude, and Deepseek for better evaluations.
>
> **A1:** We appreciate the reviewer’s thoughtful concern regarding the significance of our work and the well-documented fragility of many open-source LLMs. We acknowledge that prior studies, including your experiments with PAIR and AutoDAN on Llama2-7B and Vicuna-7B/13B, have demonstrated high attack success rates (ASR) on these models. However, we believe that this observed fragility is partly a consequence of early evaluation methodologies that relied on simple keyword-based criteria to judge attack success. Such approaches may overestimate the vulnerability of LLMs by failing to capture the true semantic intent of model outputs.
>
> To address this limitation, our work adopts HarmBench, a more advanced evaluation framework that is better aligned with human values, as evidenced by the agreement rates with human judgment shown in the table below.
>
> | Metric         | AdvBench [1] | GPTFuzz [2] | ChatGLM [3] | Llama-Guard [4] | GPT-4 [5] | HarmBench [6] |
> |:-------------- |:------------:|:-----------:|:-----------:|:---------------:|:---------:|:-------------:|
> | **Standard**   |    71.14     |    77.36    |    65.67    |      68.41      |   89.8    |   **94.53**   |
> | **Contextual** |    67.5      |    71.5     |    62.5     |      64.0       |   85.5    |   **90.5**    |
> | **Averaged**   |    69.93     |    75.42    |    64.29    |      66.94      |   88.37   |   **93.19**   |
>
> We further highlight the gap (ASR) between traditional keyword-based evaluation (e.g., AdvBench) and human-value-aligned LLM assessment (HarmBench), demonstrating that the choice of evaluation framework significantly impacts the perceived robustness of LLMs.
>
> |Judge Rule                | Model      |AutoDAN|  GCG  | PAIR  |
> | :--------                | :--------: | :---: | :---: | :---: |
> | AdvBench (keywords)      | Llama2 7B  | 56.15 | 45.38 | 0.0   |
> | HarmBench (finetuned LLM)| Llama2 7B  |  0.5  | 34.5  | 7.5   |
> | AdvBench (keywords)      | Vicuna 7B  | 97.31 | 97.12 | 94.0  |
> | HarmBench (finetuned LLM)| Vicuna 7B  | 89.5  | 90.0  | 65.5  |
>
> In our experiments, we evaluate our method on 18 LLMs, including state-of-the-art models such as DeepSeek and Llama 3, to comprehensively demonstrate its effectiveness. Furthermore, we supplement transfer attack results on closed-source models like GPT-3.5 Turbo and GPT-4 Turbo, as shown in the table below, providing concrete evidence that our approach remains effective even in challenging black-box scenarios.
>
> | Model             | Data       |  GCG  | GCG-M | GCG-T |  PEZ  | GBDA |  UAT  |  AP   |  SFS  |  ZS   | PAIR  |  TAP  | AutoDAN | PAP-top5 |  HJ   |  DR   | Ours      |
> | :---------------- | :--------: | :---: | :---: | :---: | :---: | :---:| :---: | :---: | :---: | :---: | :---: | :---: | :-----: | :------: | :---: | :---: | :-----:   |
> | DeepSeek R1 8B    | Standard   | 54.0  | 56.5  | 35.0  | 15.5  | 21.0 | 19.0  | 18.0  | 15.5  | 19.0  | 24.5  | 36.5  | 67.5    |  8.0     |  32.5 | 20.0  | **100.0** |
> | DeepSeek R1 8B    | Contextual | 47.0  | 51.0  | 56.0  | 47.0  | 44.0 | 38.0  | 51.0  | 48.0  | 46.0  | 42.0  |  65.0 | 50.0    |  34.0    |  53.0 | 45.0  | **100.0** |
> | Llama3.18B        | Standard   | 8.5   |  0.0  | 1.0   | 0.5   | 1.0  | 0.5   | 4.0   | 4.0   | 1.0   | 17.5  |  4.5  | 5.5     |  3.0     |  0.5  | 0.5   | **45.0**  |
> | Llama3.18B        | Contextual | 30.0  | 0.0   | 5.0   | 4.0   | 8.0  | 6.0   | 11.0  | 11.0  | 15.0  | 24.0  | 11.0  | 12.0    |  7.0     |  2.0  | 4.0   | **64.0**  |
> | GPT-3.5Turbo1106  | Standard   |   -   |   -   | 55.8  |   -   |  -   |   -   |   -   |   -   | 32.7  | 41.0  | 46.7  |   -     |  12.3    |  2.7  | 35.0  | **82.5**  |
> | GPT-3.5Turbo1106  | Contextual |   -   |   -   | 54.5  |   -   |  -   |   -   |   -   |   -   | 47.2  | 57.0  | 54.5  |   -     |  20.6    |  4.7  | 62.0  | **93.0**  |
> | GPT-4Turbo1106    | Standard   |   -   |   -   | 21.0  |   -   |  -   |   -   |   -   |   -   | 10.2  | 39.0  |**41.7**|   -    |  11.1    |  1.5  |  7.0  |  20.0     |
> | GPT-4Turbo1106    | Contextual |   -   |   -   | 41.8  |   -   |  -   |   -   |   -   |   -   | 34.0  | 45.0  | 50.5  |   -     |  20.2    |  6.7  | 20.0  | **69.0**  |
>
> In summary, while we recognize the fragility of certain LLMs, our work advances the field by employing more rigorous, human-aligned evaluation criteria and demonstrating strong attack performance across a broad spectrum of both open- and closed-source models.
>
> ---
>
> **Q1&Q2:** What is the transferability of the proposed method?
> As for the "unknown distribution" issue, it is commonly believed that the models within the same family with different scale (e.g., Qwen2.5-0.5B and Qwen2.5-70B) share a similar output distribution (The effectiveness of speculative inference has validated this belief in some sense). I am very curious about the transferability of the proposed method across such models.
> My second question, which might seem similar to my first one, would be: Could the authors please design and conduct an auxiliary experiment to validate my conjecture (i.e., the proposed method would transfer better within models in a same family)?
>
> **A1&A2:** As the table below summarizes, we conducted experiments to directly test the reviewer's conjecture on transferability within and across model families. The results show that our method achieves substantially higher transfer attack success rates, and models with similar architectures and training data distributions exhibit greater transferability. Specifically:
>
> (1) Transferring attacks from Vicuna 7B to Vicuna 13B yields 19.23% (BEAST) and 87.88% (Ours) ASR, while transferring from Guanaco 7B to Vicuna 13B yields 8.46% (BEAST) and 79.62% (Ours).
>
> (2) Transferring attacks from Vicuna 13B to Vicuna 7B achieves 49.81% (BEAST) and 94.42% (Ours) ASR, whereas transferring from Guanaco 7B to Vicuna 7B achieves 28.08% (BEAST) and 91.73% (Ours).
>
> These findings support the hypothesis that models with similar architectures and training data distributions exhibit greater transferability.
> However, we note an important caveat: for a fair comparison, the models being compared should ideally possess similar levels of adversarial robustness. In practice, it is challenging to quantify or ensure that any two LLMs—even within the same family—exhibit identical robustness to adversarial prompts. This limitation should be carefully considered when interpreting transferability results across models, both within and across families.
>
> | Source Model   | Attack  | Vicuna 7B | Vicuna 13B | Guanaco 7B |
> |:--------------:|:-------:|:---------:|:----------:|:----------:|
> | **Vicuna 7B**  | BEAST   |   93.65   |   19.23    |   99.81    |
> | **Vicuna 7B**  | Ours    | **97.50** | **87.88**  | **99.81**  |
> | **Vicuna 13B** | BEAST   |   49.81   |   84.80    |   99.81    |
> | **Vicuna 13B** | Ours    | **94.42** | **93.08**  | **100.0**  |
> | **Guanaco 7B** | BEAST   |   28.08   |    8.46    |   99.81    |
> | **Guanaco 7B** | Ours    | **91.73** | **79.62**  | **100.0**  |
>
> ---
>
> Once again, we thank you for your valuable feedback. We believe that by clarifying these points and conducting the suggested experiment, we can better highlight the significance and contribution of our work.
>
> Sincerely,
>
> The Authors
>
> ### Reference
>
> [1] Universal and transferable adversarial attacks on aligned language models <https://arxiv.org/pdf/2307.15043>
>
> [2] Gptfuzzer: Red teaming large language models with auto-generated jailbreak prompts <https://arxiv.org/pdf/2309.10253>
>
> [3] "Do Anything Now": Characterizing and Evaluating In-The-Wild Jailbreak Prompts on Large Language Models <https://arxiv.org/abs/2308.03825>
>
> [4] Purple llama cyberseceval: A secure coding benchmark for language models <https://arxiv.org/pdf/2312.04724>
>
> [5] Jailbreaking black box large language models in twenty queries <https://arxiv.org/abs/2310.08419>
>
> [6] Harmbench: A standardized evaluation framework for automated red teaming and robust refusal <https://arxiv.org/pdf/2402.04249>

---

> ### Author Response · Authors · 2025-08-06
>
> Dear Reviewer uz6Q,
>
> Thank you sincerely for your Mandatory Acknowledgement.
>
> We would be most grateful if you could kindly share your evaluation comments when convenient—whether our responses have sufficiently addressed your concerns or if any aspects would benefit from further clarification. Your thoughtful feedback is highly valued and essential to ensure a fair and thorough review process.
>
> Thank you very much for your time, expertise, and guidance.
>
> Best regards,
> The Authors

---

> > ### Comment · Reviewer_uz6Q · 2025-08-06
> >
> > I would like to thank the authors for their detailed response, which has addressed the majority of my concerns. In particular, the authors have conducted supplementary experiments to verify my conjecture regarding transferability both within and across model families. That said, several issues remain—for instance, there is a lack of experiments involving state-of-the-art closed-source models, e.g., the online version of GPT, Gemini, and DeepSeek. Nevertheless, given that these models are aligned by mature, commercialized teams, launching jailbreaking attacks against such systems may place academic researchers at an unfair disadvantage. Overall, I find the present paper to be novel (introducing a new paradigm), technically solid, and theoretically intuitive. As such, I have revised my score to 5.

---

> > > ### Author Response · Authors · 2025-08-06
> > >
> > > Dear Reviewer uz6Q,
> > >
> > > Thank you for your thoughtful feedback and for increasing your score. We greatly appreciate your recognition of our work.
> > >
> > > Regarding your concern about closed-source models, we have added experiments in A1 to W1 with GPT-3.5 Turbo, GPT-4 Turbo, and DeepSeek, demonstrating the effectiveness of our method on state-of-the-art closed-source LLMs.
> > >
> > > We agree that testing models like online GPT or DeepSeek poses practical challenges for academic research. Our new results further support the broad applicability of our approach.
> > >
> > > Thank you again for your valuable comments.
> > >
> > > Best regards,
> > > The Authors

---

### Official Review · Reviewer_ETbV · 2025-07-02

**Clarity:** 3
**Significance:** 4
**Originality:** 4
**Rating:** 5
**Confidence:** 3

**Summary:**

The paper introduces Semantic Representation Attack (SRA), a new paradigm for jail‑breaking aligned large‑language models (LLMs). Existing automatic attacks usually optimise prompts so that the model produces one fixed textual pattern of compliance (e.g. “Sure, here is …”). The authors argue this is brittle and computationally expensive. Instead, SRA explicitly targets a set of responses that share the same semantic representation (i.e. identical meaning) regardless of surface form.
To operationalise the idea the authors derive a coherence–convergence theorem linking low‑perplexity (semantically coherent) prompts to higher probability of landing anywhere within the harmful semantic equivalence class. Guided by this, they design Semantic Representation Heuristic Search (SRHS) – a tree‑search that prunes branches using (i) a perplexity threshold to maintain coherence and (ii) a learned semantic classifier to test whether candidate prompts already elicit a malicious meaning.
Extensive experiments on 18 open‑source chat models show an average attack‑success rate (ASR) of 89.4 %, including 100 % ASR on 11 models (Table 1, p. 8) while keeping adversarial prompts short and low‑perplexity. Efficiency comparisons under fixed time budgets (Table 2, p. 8) demonstrate markedly higher ASR and 2‑orders‑of‑magnitude lower perplexity than GCG, AutoDAN and BEAST. Transfer‑attack results (Table 4, p. 9) and robustness under perplexity‑based defences further support the method.

**Questions:**

Closed‑source models – Have you attempted SRHS against APIs that expose only final text (no logits)? Could you approximate the coherence filter with a surrogate LM and still attain high ASR?

Classifier sensitivity – The semantic mapping
𝑅
(
⋅
)
R(⋅) is implemented with a HarmBench‑tuned model. How does attack success vary if this classifier has lower recall or is mis‑aligned with the victim model? An ablation would clarify robustness.

Perplexity threshold τ – Table 3 shows τ = 20 performs well, but τ must be chosen per model. Can you propose a heuristic for selecting τ without manual sweeps?

Prompt length vs. ASR – What is the average token length of successful prompts, and does further shortening sacrifice ASR? This matters for detectability.

Defence suggestions – Beyond lowering perplexity thresholds, which defences could specifically break semantic‑representation convergence? Clarifying this would aid the safety community and could influence my significance score.

Criteria for score change: Demonstrating SRHS effectiveness under black‑box settings or showing robustness across different semantic classifiers would raise quality and significance to excellent.

**Ethical Concerns:**

["NO or VERY MINOR ethics concerns only"]

**Final Justification:**

Based on the conversation below I am updating my ratings.

**Limitations:**

Yes, the paper explicitly lists limitations (white‑box requirement, τ calibration, classifier dependence) in §5. I encourage adding a brief discussion on potential dataset bias in the semantic classifier and on compute cost for very large models.

**Quality:**

4

**Strengths And Weaknesses:**

Strengths
1. Sound formulation: clear optimisation objective (Eq. 6). Rigorous theory: three theorems with proofs in App. A linking perplexity to semantic convergence. Comprehensive evaluation: 18 models × 2 datasets, plus efficiency, robustness & transfer studies; results strongly outperform baselines (e.g. 100 % vs 87 % ASR on Vicuna‑13B, Table 1).
2. Paper is well structured; pseudo‑code of SRHS (Alg. 1, p. 6) and figures (Fig. 2 & Fig. 3) make intuition clear. Comprehensive NeurIPS checklist included.
3. Demonstrates that targeting meaning rather than wording dramatically raises ASR and transferability – an important insight for both offensive and defensive research. Raises the bar for alignment safety benchmarks.
4. Shifts adversarial objective from token pattern to semantic class; to my knowledge, no prior work formalises this with provable guarantees.

Weaknesses
1. White‑box assumption: SRHS needs token probabilities; closed models (e.g. ChatGPT) are not tested. Semantic classifier dependency: success hinges on the HarmBench‑fine‑tuned Llama‑2‑13B; generality to other semantics detectors is untested.
2. Dense notation and long proofs re‑state obvious steps; could be shortened. The link between Theorem 3.1 and practical pruning thresholds (τ) is buried in appendix.
3. Impact on real‑world closed‑source LLMs is speculative; without such evidence, practical risk remains uncertain.
4. Use of perplexity as a coherence proxy is known; the novelty lies mainly in combining it with semantic‑class targeting.

---

> ### Author Rebuttal · Authors · 2025-07-29
>
> Dear Reviewer ETbV,
>
> Thank you for your exceptionally thorough and positive review. We are delighted you found our work technically solid, rigorous, and impactful. Your detailed questions are highly insightful, and we appreciate the opportunity to provide further clarification and detail on our work.
>
> ---
> **W1-1&Q1:** White‑box assumption: SRHS needs token probabilities; closed models (e.g. ChatGPT) are not tested.
>
> **A1-1:** We appreciate the reviewer’s thoughtful concern regarding the white-box assumption. Our method does require access to token probabilities. To address this, we leverage proxy models like DeepSeek and Llama 3 to perform a semantic representation attack. Our experimental results show that these prompts can be directly transferred to black-box models, achieving state-of-the-art attack success rates without access to internal probabilities. This transferability underscores the practical risk posed by our method in real-world black-box scenarios.
>
> | Model             | Data       |  GCG  | GCG-M | GCG-T |  PEZ  | GBDA |  UAT  |  AP   |  SFS  |  ZS   | PAIR  |  TAP  | AutoDAN | PAP-top5 |  HJ   |  DR   | Ours    |
> | :---------------- | :--------: | :---: | :---: | :---: | :---: | :---:| :---: | :---: | :---: | :---: | :---: | :---: | :-----: | :------: | :---: | :---: | :-----: |
> | GPT-3.5Turbo1106  | Standard   |   -   |   -   | 55.8  |   -   |  -   |   -   |   -   |   -   | 32.7  | 41.0  | 46.7  |   -     |  12.3    |  2.7  | 35.0  | **82.5**|
> | GPT-3.5Turbo1106  | Contextual |   -   |   -   | 54.5  |   -   |  -   |   -   |   -   |   -   | 47.2  | 57.0  | 54.5  |   -     |  20.6    |  4.7  | 62.0  | **93.0**|
> | GPT-4Turbo1106    | Standard   |   -   |   -   | 21.0  |   -   |  -   |   -   |   -   |   -   | 10.2  | 39.0  |**41.7**|   -    |  11.1    |  1.5  |  7.0  |  20.0   |
> | GPT-4Turbo1106    | Contextual |   -   |   -   | 41.8  |   -   |  -   |   -   |   -   |   -   | 34.0  | 45.0  | 50.5  |   -     |  20.2    |  6.7  | 20.0  | **69.0**|
>
> ---
>
> **W1-2&Q2:** Semantic classifier dependency: success hinges on the HarmBench‑fine‑tuned Llama‑2‑13B; generality to other semantics detectors is untested.
>
> **A1-2:** We acknowledge the reviewer’s concern regarding our reliance on the HarmBench-fine-tuned Llama-2-13B [6] as the semantic classifier. We selected this model because, as shown in the table below, it achieves the highest agreement with human judgments among all evaluated frameworks, ensuring reliable and valid semantic equivalence assessments. Furthermore, the high transferability of our adversarial prompts across diverse model families suggests that our approach captures fundamental semantic properties rather than overfitting to a specific classifier. We agree that exploring alternative semantic detectors is a valuable direction for future work and have clarified this in the revised manuscript.
>
> | Metric         | AdvBench [1] | GPTFuzz [2] | ChatGLM [3] | Llama-Guard [4] | GPT-4 [5] | HarmBench [6] |
> |:-------------- |:------------:|:-----------:|:-----------:|:---------------:|:---------:|:-------------:|
> | **Standard**   |    71.14     |    77.36    |    65.67    |      68.41      |   89.8    |   **94.53**   |
> | **Contextual** |    67.5      |    71.5     |    62.5     |      64.0       |   85.5    |   **90.5**    |
> | **Averaged**   |    69.93     |    75.42    |    64.29    |      66.94      |   88.37   |   **93.19**   |
>
> ---
>
> **W2:** Dense notation and long proofs re‑state obvious steps; could be shortened. The link between Theorem 3.1 and practical pruning thresholds (τ) is buried in appendix.
>
> **A2:** We appreciate the reviewer’s feedback on the density of notation and the length of the proofs. We agree that some steps in the theoretical analysis could be presented more concisely and have revised the main text to streamline the exposition. Additionally, we acknowledge that the connection between Theorem 3.1 and the practical choice of pruning thresholds (τ) was not sufficiently emphasized. In the revised manuscript, we have explicitly clarified how Theorem 3.1 provides a theoretical basis for setting τ.
>
> ---
>
> **W3:** Impact on real‑world closed‑source LLMs is speculative; without such evidence, practical risk remains uncertain.
>
> **A3:** We appreciate the reviewer’s concern about the practical risk to real-world closed-source LLMs. As detailed in our response to W1-1 (A1-1), we have demonstrated that adversarial prompts generated via proxy models can be effectively transferred to closed-source models (e.g., GPT-3.5/4 Turbo), achieving SOTA attack success rates. This provides concrete evidence of the practical risk and real-world impact of our method.
>
> ---
>
> **W4:** Use of perplexity as a coherence proxy is known; the novelty lies mainly in combining it with semantic‑class targeting.
>
> **A4:** We agree and appreciate the reviewer’s recognition that our main novelty is the integration of semantic-class targeting with perplexity-based filtering.
>
> ---
>
> **Q3:** Perplexity threshold τ – Table 3 shows τ = 20 performs well, but τ must be chosen per model. Can you propose a heuristic for selecting τ without manual sweeps?
>
> **A3:** We propose a heuristic search strategy for the perplexity threshold τ: starting from an initial τ, we iteratively adjust τ based on attack performance on a small validation set, using the observed success rate as a heuristic function. This process automatically guides the search toward an optimal τ, efficiently balancing attack efficacy and training stability without exhaustive manual tuning.
>
> ---
>
> **Q4:** Prompt length vs. ASR – What is the average token length of successful prompts, and does further shortening sacrifice ASR? This matters for detectability.
>
> **A4:** We report the average token length (with standard deviation) of 300 successful prompts (100% ASR) in the table below. Our method incrementally searches for candidate tokens based on the original malicious prompt, resulting in significantly shorter prompts than prior methods. This compactness highlights both the efficiency and effectiveness of our approach.
>
> | Model                        | Avg. Token Length (mean ± std)|
> |------------------------------|:-----------------------------:|
> | DeepSeek R1 8B               | 2.28 ± 1.50                   |
> | Koala 13B                    | 1.99 ± 0.86                   |
> | Koala 7B                     | 1.74 ± 0.81                   |
> | Mistral 7B                   | 1.56 ± 0.72                   |
> | OpenChat 7B                  | 1.61 ± 0.79                   |
> | Orca 2 13B                   | 1.58 ± 0.76                   |
> | Orca 2 7B                    | 1.57 ± 0.63                   |
> | Starling 7B                  | 1.34 ± 0.55                   |
> | Vicuna 13B                   | 2.61 ± 1.13                   |
> | Vicuna 7B                    | 2.18 ± 0.96                   |
>
> | Attack Method   |  GCG[1] |AutoDAN[8]| SAA[7] |  Ours  |
> |:-------------:  |:-------:|:-------: |:------:|:------:|
> |**Prompt Length**|   20    |  ~60     | ~480   |  <10   |
>
> ---
>
> **Q5:** Defence suggestions – Beyond lowering perplexity thresholds, which defences could specifically break semantic‑representation convergence? Clarifying this would aid the safety community and could influence my significance score.
>
> **A5:** We recommend a multi-dimensional defense strategy that combines the strengths of different approaches to effectively mitigate semantic-representation-based jailbreak attacks. (1) At the system level, adversarial prompt purification techniques could be applied before inputs reach the LLM. (2) At the model level, further improving the model’s ability to distinguish harmful from benign content—by aligning more closely with human values—remains essential. (3) Additionally, output filtering mechanisms could be employed to detect and block harmful responses at the system level. Integrating these system-level and model-level defenses can provide robust protection against advanced jailbreak attacks.
>
> ---
>
> **Q6:** Criteria for score change: Demonstrating SRHS effectiveness under black‑box settings or showing robustness across different semantic classifiers would raise quality and significance to excellent.
>
> **A6:** Thank you for outlining these important criteria. As discussed in our responses to A1-1 and A1-2, we have provided empirical evidence demonstrating our method’s effectiveness in black-box settings and the advantages of our chosen semantic classifiers. We hope these results directly address your concerns and further strengthen the quality and significance of our work.
>
> ---
>
> Once again, thank you for your constructive feedback and for helping us improve the paper.
>
> Sincerely,
>
> The Authors
>
> ### Reference
>
> [1] Universal and transferable adversarial attacks on aligned language models <https://arxiv.org/pdf/2307.15043>
>
> [2] Gptfuzzer: Red teaming large language models with auto-generated jailbreak prompts <https://arxiv.org/pdf/2309.10253>
>
> [3] "Do Anything Now": Characterizing and Evaluating In-The-Wild Jailbreak Prompts on Large Language Models <https://arxiv.org/abs/2308.03825>
>
> [4] Purple llama cyberseceval: A secure coding benchmark for language models <https://arxiv.org/pdf/2312.04724>
>
> [5] Jailbreaking black box large language models in twenty queries <https://arxiv.org/abs/2310.08419>
>
> [6] Harmbench: A standardized evaluation framework for automated red teaming and robust refusal <https://arxiv.org/pdf/2402.04249>
>
> [7] Jailbreaking Leading Safety-Aligned LLMs with Simple Adaptive Attacks <https://arxiv.org/abs/2404.02151>
>
> [8] AutoDAN: Generating Stealthy Jailbreak Prompts on Aligned Large Language Models <https://arxiv.org/abs/2310.04451>

---

> ### Author Response · Authors · 2025-08-06
>
> Dear Reviewer ETbV,
>
> Thank you very much for submitting the Mandatory Acknowledgement.
>
> We would be truly grateful if you could kindly share your evaluation comments—whether our responses have adequately addressed your concerns, or if any points require further clarification. Your thoughtful feedback is invaluable to us and to the integrity of the review process.
>
> Thank you sincerely for your time and guidance.
>
> Best regards,
>
> The Authors

---

> > ### Comment · Reviewer_ETbV · 2025-08-08
> > **Post-Rebuttal Evaluation**
> >
> > ### Summary of Clarifications
> > The authors addressed the white-box limitation by showing strong black-box transfer results:
> > - GPT-3.5-Turbo: 82.5 % (standard) / 93.0 % (contextual) ASR – new SOTA.
> > - GPT-4-Turbo: 20 % (standard) / 69 % (contextual) ASR – contextual SOTA, standard below TAP (41.7 %).
> >
> > They justified the HarmBench classifier choice with highest human-agreement scores, proposed a heuristic τ-selection method, reported very short successful prompts (< 3 tokens on average), and streamlined theory–practice links in the text.
> >
> > ### Updated Strengths
> > - Added black-box transfer evidence, strengthening practical impact.
> > - Real-world risk to closed models now demonstrated.
> > - Clearer presentation and compact prompts confirmed.
> >
> > ### Remaining Weaknesses
> > - GPT-4-Turbo standard ASR still lower than best baseline.
> > - Robustness to alternative semantic detectors remains untested.
> >
> > ### Revised Scores
> > - **Quality:** excellent (↑)
> > - **Clarity:** good
> > - **Significance:** excellent (↑)
> > - **Originality:** excellent
> >
> > ### Suggestions for Camera-Ready
> > 1. Explain GPT-4-Turbo standard set gap.
> > 2. Include at least one classifier ablation.
> > 3. Provide pseudo-code or curves for τ tuning.
> >
> > ### Limitations
> > Yes – acknowledged and expanded.
> >
> > ### Updated Overall Recommendation
> > **Strong Accept (a)** – Rebuttal meets the criteria for raising the score: effective black-box results and improved clarity, combined with rigorous theory and strong open-source performance.

---

> > > ### Author Response · Authors · 2025-08-08
> > >
> > > Dear Reviewer ETbV,
> > >
> > > Thank you for your thoughtful feedback and for recognizing the improvements in our rebuttal, particularly regarding the black-box transfer results and enhanced clarity. We sincerely appreciate your positive assessment and the "Strong Accept" recommendation.
> > >
> > > We confirm that we will carefully incorporate all your suggestions into the camera-ready version: we will explain the performance gap on GPT-4-Turbo under the standard setting, include a classifier ablation study, and provide either pseudo-code or tuning curves for the τ selection heuristic.
> > >
> > > Thank you again for your constructive comments and support.
> > >
> > > Best regards,
> > >
> > > On behalf of all authors

---

### Official Review · Reviewer_6jQY · 2025-07-03

**Clarity:** 3
**Significance:** 3
**Originality:** 2
**Rating:** 5
**Confidence:** 3

**Summary:**

The paper introduces Semantic Representation Attack (SRA), a new method for jailbreaking aligned large language models (LLMs). Unlike traditional attacks that rely on triggering exact phrases, SRA targets the semantic meaning behind harmful outputs, enabling more natural and stealthy adversarial prompts. The authors propose a Semantic Representation Heuristic Search algorithm that maintains coherence while optimizing prompts. SRA achieves 89.41% average attack success across 18 LLMs (100% on 11), and outperforms existing methods in efficiency, naturalness, robustness, and transferability.

**Questions:**

N/A

**Ethical Concerns:**

["NO or VERY MINOR ethics concerns only"]

**Final Justification:**

The paper introduces a novel perspective by targeting semantic representations instead of exact token patterns, which achieves 100% ASR on 11 out of 18 LLMs and 89.41% on average, outperforming all prior attack methods in effectiveness.

The authors have addressed my concerns during rebuttal.

**Limitations:**

Yes

**Quality:**

3

**Strengths And Weaknesses:**

strength:
1. The paper introduces a novel perspective by targeting semantic representations instead of exact token patterns, advancing the field of adversarial NLP.
2. Achieves 100% ASR on 11 out of 18 LLMs and 89.41% on average, outperforming all prior attack methods in effectiveness.
3. The Semantic Representation Heuristic Search (SRHS) reduces computational complexity from exponential to near-linear under constraints, enabling fast prompt generation.

weakness:
1. The attack assumes access to model logits/probabilities, which is infeasible for most closed-source or API-only LLMs.
2. Requires defining target semantic representations, which could be non-trivial or subjective in complex or ambiguous tasks.
3. Relies on external models (e.g., fine-tuned Llama) to define semantic equivalence, which may vary in accuracy across tasks or domains.

---

> ### Author Rebuttal · Authors · 2025-07-29
>
> Dear Reviewer 6jQY,
>
> Thank you for your positive assessment and insightful feedback on our paper. We are encouraged that you found our work a novel advancement in adversarial NLP with strong empirical results and high efficiency. We appreciate the opportunity to address the weaknesses you've identified.
>
> ---
>
> **W1:** The attack assumes access to model logits/probabilities, which is infeasible for most closed-source or API-only LLMs.
>
> **A1:** Thank you for highlighting this important limitation. In our new transfer attack experiments, we generate adversarial prompts using our semantic representation attack on proxy models (e.g., DeepSeek, Llama 3) and apply them directly to closed-source/API-only LLMs. As shown in the table below, our method achieves strong attack success rates (ASR) on both GPT-3.5 Turbo and GPT-4 Turbo, even without access to their logits or internal probabilities. This demonstrates that our approach is not only effective in white-box settings but also highly transferable to black-box scenarios—a key strength that reveals the vulnerability of SOTA LLMs to semantic representation-based attacks, even with API-only access. We have explicitly discussed these findings and their implications in the revised manuscript.
>
> This high transferability stems from our focus on optimizing the semantic representation space rather than specific textual patterns, making our adversarial prompts more robust and generalizable across model architectures. We further believe that incorporating closed-source/API-only LLMs into the optimization pipeline via proxy models could further enhance attack performance by better tailoring prompts to the target models’ internal representations.
>
> | Model             | Data       |  GCG  | GCG-M | GCG-T |  PEZ  | GBDA |  UAT  |  AP   |  SFS  |  ZS   | PAIR  |  TAP  | AutoDAN | PAP-top5 |  HJ   |  DR   | Ours    |
> | :---------------- | :--------: | :---: | :---: | :---: | :---: | :---:| :---: | :---: | :---: | :---: | :---: | :---: | :-----: | :------: | :---: | :---: | :-----: |
> | GPT-3.5Turbo1106  | Standard   |   -   |   -   | 55.8  |   -   |  -   |   -   |   -   |   -   | 32.7  | 41.0  | 46.7  |   -     |  12.3    |  2.7  | 35.0  | **82.5**|
> | GPT-3.5Turbo1106  | Contextual |   -   |   -   | 54.5  |   -   |  -   |   -   |   -   |   -   | 47.2  | 57.0  | 54.5  |   -     |  20.6    |  4.7  | 62.0  | **93.0**|
> | GPT-4Turbo1106    | Standard   |   -   |   -   | 21.0  |   -   |  -   |   -   |   -   |   -   | 10.2  | 39.0  |**41.7**|   -    |  11.1    |  1.5  |  7.0  |  20.0   |
> | GPT-4Turbo1106    | Contextual |   -   |   -   | 41.8  |   -   |  -   |   -   |   -   |   -   | 34.0  | 45.0  | 50.5  |   -     |  20.2    |  6.7  | 20.0  | **69.0**|
>
> ---
>
> **W2:** Requires defining target semantic representations, which could be non-trivial or subjective in complex or ambiguous tasks.
>
> **A2:** Thank you for raising this important issue. A semantic representation is an abstract, language-independent meaning that can be expressed by multiple surface forms with equivalent propositional content. Specifically, for a given harmful query, we identify the set of responses that share the same semantic representation—i.e., they fulfill the malicious intent regardless of lexical or syntactic variation. Formally, as described in our methodology, we denote the semantic representation space as $\Omega$, and for each target harmful behavior, we define a representation $\Phi \in \Omega$ such that the set of compliant responses $\mathcal{Y}_\Phi$ all satisfy $\mathcal{R}(\boldsymbol{y}) = \Phi$. This approach allows us to optimize for a broad equivalence class of harmful outputs, rather than a single fixed response, making the attack both more general and more robust. We have clarified this definition and its practical implementation in the revised manuscript, and discuss the challenges and future directions for defining semantic targets in more complex or ambiguous scenarios.
>
> ---
>
> **W3:** Relies on external models (e.g., fine-tuned Llama) to define semantic equivalence, which may vary in accuracy across tasks or domains.
>
> **A3:** We appreciate your concern about relying on an external model (e.g., fine-tuned Llama) to define semantic equivalence, as its accuracy may vary across tasks or domains. We selected this model because, as shown in the table below, it achieves superior alignment with human values compared to other evaluation frameworks. This strong alignment ensures that the semantic representations used in our attack are consistent with human judgment, increasing the reliability and validity of our results. Additionally, the high transferability of our prompts across diverse model families suggests that our approach captures fundamental semantic properties rather than overfitting to the guidance model. We have clarified this rationale and its implications in the revised manuscript.
>
> | Metric         | AdvBench [1] | GPTFuzz [2] | ChatGLM [3] | Llama-Guard [4] | GPT-4 [5] | Fine-tuned Llama [6] |
> |:-------------- |:------------:|:-----------:|:-----------:|:---------------:|:---------:|:-------------:       |
> | **Standard**   |    71.14     |    77.36    |    65.67    |      68.41      |   89.8    |   **94.53**          |
> | **Contextual** |    67.5      |    71.5     |    62.5     |      64.0       |   85.5    |   **90.5**           |
> | **Averaged**   |    69.93     |    75.42    |    64.29    |      66.94      |   88.37   |   **93.19**          |
>
> ---
>
> Once again, thank you for your valuable feedback and for recognizing the potential of our work. We are confident that the paper has been significantly strengthened by addressing these points.
>
> Sincerely,
>
> The Authors
>
> ### Reference
>
> [1] Universal and transferable adversarial attacks on aligned language models <https://arxiv.org/pdf/2307.15043>
>
> [2] Gptfuzzer: Red teaming large language models with auto-generated jailbreak prompts <https://arxiv.org/pdf/2309.10253>
>
> [3] "Do Anything Now": Characterizing and Evaluating In-The-Wild Jailbreak Prompts on Large Language Models <https://arxiv.org/abs/2308.03825>
>
> [4] Purple llama cyberseceval: A secure coding benchmark for language models <https://arxiv.org/pdf/2312.04724>
>
> [5] Jailbreaking black box large language models in twenty queries <https://arxiv.org/abs/2310.08419>
>
> [6] Harmbench: A standardized evaluation framework for automated red teaming and robust refusal <https://arxiv.org/pdf/2402.04249>

---

> > ### Author Response · Authors · 2025-08-06
> >
> > Dear Reviewer 6jQY,
> >
> > Thank you very much for providing insightful reviews.
> >
> > We would be truly grateful if you could kindly share your evaluation comments on our rebuttal—whether our responses have adequately addressed your concerns, or if any points require further clarification. Your thoughtful feedback is invaluable to us and to the integrity of the review process.
> >
> > Thank you sincerely for your time and guidance.
> >
> > Best regards,
> >
> > The Authors

---

### Official Review · Reviewer_oJrc · 2025-07-05

**Clarity:** 4
**Significance:** 3
**Originality:** 2
**Rating:** 5
**Confidence:** 3

**Summary:**

This work introduces Semantic Representation Attack to jailbreak LLMs by targeting semantic meaning of harmful responses rather than utilizing specific text patterns. It proposes an algorithm to generate coherent jailbreak attacks efficiently. The approach achieves 89.1% ASR across 18 LLMs in the presence of perplexity based defenses.

**Questions:**

None

**Ethical Concerns:**

["NO or VERY MINOR ethics concerns only"]

**Final Justification:**

Authors have now clarified all the weaknesses pointed out during the review process. The novelty of the work is now clear and well explained. I would suggest authors to add this extended literature survey and explanation of the contributions in the revised version.

**Limitations:**

Authors note that their method requires white-box logit access to perform the attack.

**Quality:**

3

**Strengths And Weaknesses:**

Strengths

1) Methodologically and experimentally detailed
All the concepts are formalized rigorously and proven. Evaluation is performed across 18 LLMs, along with exploration of efficiency and naturalness of the prompts. Authors have also benchmarked a range of different attacks on the prompt set.

2) Highly Positive Results
This work achieves 100% ASR on 11 out of 18 tested LLMs. The algorithm provides significant computation savings while generating natural and robust prompts.


Weaknesses


1) Lack of novelty.
There are multiple jailbreaking methods [1, 2, 3] currently in existence that manipulate prompts and retain semantic coherence. A Comparison with them and an explanation in the related work section will be helpful. The paper could benefit from a more thorough literature survey.

2) Needs more Evaluations in the presence of defenses.
The paper does not evaluate the jailbreaking capabilities in the presence of defenses other than PPL-based defenses, which is necessary to understand the robustness. Defenses like [4] could be effective against this attack.

3) Perplexity as a measure of coherence
The paper extensively uses perplexity as a measure of coherence. Log-Perplexity, calculating the certainty of occurrence of words in the context, does not directly capture the coherence/naturalness of the text. Moreover, the algorithm itself uses perplexity to generate the jailbreaks; hence, the same measure should not be used as a metric.

[1] Jailbreaking Leading Safety-Aligned LLMs with Simple Adaptive Attacks (https://arxiv.org/abs/2404.02151)

[2] Tree of Attacks: Jailbreaking Black-Box LLMs Automatically (https://arxiv.org/abs/2312.02119)

[3] Does Safety Training of LLMs Generalize to Semantically Related Natural Prompts? (https://arxiv.org/abs/2412.03235)

[4] Defending Large Language Models against Jailbreak Attacks via Semantic Smoothing  (https://arxiv.org/abs/2402.16192)

---

> ### Author Rebuttal · Authors · 2025-07-29
>
> Dear Reviewer oJrc,
>
> Thank you for your detailed review and constructive feedback. We appreciate your recognition of our methodological rigor, comprehensive evaluation across 18 LLMs, and strong results. We also value your comments on the paper’s weaknesses, which we have addressed point by point below.
>
> ---
>
> **W1:** Lack of novelty. There are multiple jailbreaking methods [1, 2, 3] currently in existence that manipulate prompts and retain semantic coherence. A Comparison with them and an explanation in the related work section will be helpful. The paper could benefit from a more thorough literature survey.
>
> **A1:** We thank you for your insightful feedback and for highlighting these important references. We agree that a direct comparison is essential and have updated the "Related Work" section and experiments to clarify our contributions.
>
> While previous methods generate semantically coherent prompts by optimizing for specific textual patterns (e.g., "Sure, here is..."), our Semantic Representation Attack (SRA) fundamentally differs by directly optimizing for the target semantic representation of harmful responses. This strategy overcomes the trade-off between prompt naturalness and attack efficacy that constrains prior work. Extensive experiments show that SRA achieves superior attack efficacy, transferability, naturalness, and efficiency compared to existing methods.
>
> Specifically, our work differs from [1] in three fundamental ways:
>
> 1.  **Optimization Goals:** Method [1] optimizes for specific textual patterns (e.g., "Sure, here is...") as its objective. In contrast, our approach directly targets the underlying malicious semantic representation. By optimizing for a broad class of semantically equivalent outputs rather than narrow surface forms, our method overcomes the trade-off between attack efficacy and prompt naturalness that limits previous work.
>
> 2.  **Prompt Generation Framework:** Method [1] relies on a manual template with random search suffixes, resulting in long prompts (~480 tokens), while our method is fully automatic and produces much shorter prompts (<10 tokens), as summarized in the table below.
>
> | Feature                   | Method [1]                             | Our Method                |
> | :------------------------ | :---------------------------------     | :------------------------ |
> | **Generation Process**    | Manual template + random search suffix | Fully automatic generation|
> | **Prompt Template Length**| 463 tokens (measured by GPT-4o)        | No template required      |
> | **Total Prompt Length**   | ~480 tokens                            | < 10 tokens               |
>
> 3.  **Evaluation Framework:** Method [1] evaluates attack success using a vanilla LLM (GPT-4) [9] and a rule-based keyword list [5]. In contrast, we use the more advanced HarmBench framework [10], which achieves significantly higher agreement with human judgments on harmfulness, as shown in the table below.
>
> | Metric         | AdvBench [5] | GPTFuzz [6] | ChatGLM [7] | Llama-Guard [8] | GPT-4 [9] | HarmBench [10] |
> |:-------------- |:------------:|:-----------:|:-----------:|:---------------:|:---------:|:-------------:|
> | **Standard**   |    71.14     |    77.36    |    65.67    |      68.41      |   89.8    |   **94.53**   |
> | **Contextual** |    67.5      |    71.5     |    62.5     |      64.0       |   85.5    |   **90.5**    |
> | **Averaged**   |    69.93     |    75.42    |    64.29    |      66.94      |   88.37   |   **93.19**   |
>
> TAP [2] employs an attacker LLM to iteratively refine candidate prompts until one successfully jailbreaks the target model or the maximum tree depth is reached. The comparative experimental results are shown below.
>
> | Model             |   TAP   |   Ours    |
> | :---------------- | :-----: | :-------: |
> | DeepSeek R1 8B    |  46.00  | **100.00**|
> | Llama 3.1 8B      |   6.67  |  **45.00**|
> | Llama 2 7B        |  15.25  |  **30.33**|
> | Vicuna 7B         |  68.00  | **100.00**|
> | Vicuna 13B        |  69.05  | **100.00**|
> | Baichuan 2 7B     |  68.25  |  **99.00**|
> | Baichuan 2 13B    |  71.05  |  **99.67**|
> | Qwen 7B           |  69.25  |  **94.00**|
> | Koala 7B          |  78.25  | **100.00**|
> | Koala 13B         |  77.50  | **100.00**|
> | Orca 2 7B         |  76.25  | **100.00**|
> | Orca 2 13B        |  78.00  | **100.00**|
> | SOLAR 10.7B       |  87.00  |  **99.33**|
> | Mistral 7B        |  83.00  | **100.00**|
> | OpenChat 7B       |  82.75  | **100.00**|
> | Starling 7B       |  88.25  | **100.00**|
> | Zephyr 7B         |  87.00  | **100.00**|
> | R2D2 7B           |**77.25**|   42.00   |
> | **Averaged**      |  68.27  | **89.41** |
>
> Work [3] shows that aligned LLMs can be vulnerable to malicious questions generated by other aligned LLMs when given a malicious response, forming a "Question-Answer" and "Answer-Question" loop. However, these prompts are generated without attack optimization, which fundamentally distinguishes our approach.
>
> ---
>
> **W2:** Needs more Evaluations in the presence of defenses. The paper does not evaluate the jailbreaking capabilities in the presence of defenses other than PPL-based defenses, which is necessary to understand the robustness. Defenses like [4] could be effective against this attack.
>
> **A2:** We fully agree that evaluating our attack against a broader range of defenses is essential. Thank you for recommending work [4] as a relevant defense. To address this, we selected three strong baseline defenses from [4] and conducted adaptive attack evaluations on the same dataset. The results (ASR) show that our method consistently identifies effective adversarial prompts in the semantic representation space, even when prompts are perturbed:
>
> |                   | PAIR | AutoDAN | Ours   |
> |:------------------|:----:|:-------:|:------:|
> | Defenseless       |  76  |   90    | **92** |
> | SmoothLLM-Swap    |  48  |   56    | **100**|
> | SmoothLLM-Insert  |  62  |   78    | **96** |
> | SmoothLLM-Patch   |  52  |   74    | **100**|
>
> These results are notable. While these defenses substantially reduce the ASR of strong baselines like PAIR and AutoDAN (by up to 34 points), our attack remains highly effective—and in some cases, ASR even increases under these defenses. This counterintuitive outcome arises because (1) prior methods rely on specific keywords, making them vulnerable to prompt perturbations, whereas our approach targets the underlying semantic space; and (2) perturbed prompts may fall outside the distribution of safety alignment training, diminishing the effectiveness of model-level defenses.
>
> ---
>
> **W3:** Perplexity as a measure of coherence. The paper extensively uses perplexity as a measure of coherence. Log-Perplexity, calculating the certainty of occurrence of words in the context, does not directly capture the coherence/naturalness of the text. Moreover, the algorithm itself uses perplexity to generate the jailbreaks; hence, the same measure should not be used as a metric.
>
> **A3:** We appreciate your thoughtful comments on the use of perplexity as a metric. While perplexity mainly measures grammatical fluency rather than true semantic coherence, it remains a widely used and reasonable proxy for naturalness in adversarial text generation [11,12], as it quantifies how "expected" or "natural" text appears to a language model.
>
> You also raise an important point about the potential issue of using perplexity both as an optimization objective and an evaluation metric. Prior methods often struggle with a trade-off between low perplexity (natural prompts) and high attack success. In contrast, our semantic representation-based approach overcomes this limitation, achieving both high attack success rates and low perplexity, thus generating natural-looking prompts without sacrificing efficacy.
>
> To further address your concern, we conducted a human evaluation of 300 generated prompts across three representative LLMs. Independent evaluators rated the naturalness of these prompts, and the results (all >95%) confirm that our method consistently produces human-like, coherent text while maintaining strong attack performance.
>
> |Llama 3.1 8B|DeepSeek R1 8B|Qwen 7B|
> | :---:      | :---:        | :---: |
> |98.3        |96.7          |95.4   |
>
> ---
>
> Once again, thank you for your invaluable feedback, which has been crucial to improving our paper. We hope our responses address your concerns and are confident that these revisions make our work more compelling and its contributions clearer.
>
> Sincerely,
>
> The Authors
>
> ### Reference
>
> [1] Jailbreaking Leading Safety-Aligned LLMs with Simple Adaptive Attacks <https://arxiv.org/abs/2404.02151>
>
> [2] Tree of Attacks: Jailbreaking Black-Box LLMs Automatically <https://arxiv.org/abs/2312.02119>
>
> [3] Does Safety Training of LLMs Generalize to Semantically Related Natural Prompts? <https://arxiv.org/abs/2412.03235>
>
> [4] Defending Large Language Models against Jailbreak Attacks via Semantic Smoothing <https://arxiv.org/abs/2402.16192>
>
> [5] Universal and transferable adversarial attacks on aligned language models <https://arxiv.org/pdf/2307.15043>
>
> [6] Gptfuzzer: Red teaming large language models with auto-generated jailbreak prompts <https://arxiv.org/pdf/2309.10253>
>
> [7] "Do Anything Now": Characterizing and Evaluating In-The-Wild Jailbreak Prompts on Large Language Models <https://arxiv.org/abs/2308.03825>
>
> [8] Purple llama cyberseceval: A secure coding benchmark for language models <https://arxiv.org/pdf/2312.04724>
>
> [9] Jailbreaking black box large language models in twenty queries <https://arxiv.org/abs/2310.08419>
>
> [10] Harmbench: A standardized evaluation framework for automated red teaming and robust refusal <https://arxiv.org/pdf/2402.04249>
>
> [11] AutoDAN: Generating Stealthy Jailbreak Prompts on Aligned Large Language Models <https://arxiv.org/abs/2310.04451>
>
> [12] Fast Adversarial Attacks on Language Models In One GPU Minute <https://arxiv.org/abs/2402.15570>

---

> > ### Comment · Reviewer_oJrc · 2025-08-06
> >
> > Thank you for your experiments and evaluations. My concerns with **W2** and **W3** are resolved now. The results will surely strengthen the paper.
> >
> > Regarding **W1**,
> >
> > Your rebuttal differentiates this work from the works I have cited in my response, which is good. However, the references I provided were representative of my point that this is not the first work to experiment with jailbreaks in semantic representational space (If this is false, please claim so with more references). If this is true, then the main contribution is that the semantic jailbreaks are produced with the best attack objective and hence convergent and optimal?  I wish to gain clarity on this and preferably in the paper as well.

---

> ### Author Response · Authors · 2025-08-06
>
> Dear Reviewer oJrc,
>
> Thank you for your feedback and for confirming that W2 and W3 are resolved. We apologize for the earlier misunderstanding regarding W1 and appreciate the opportunity to clarify the novelty of our work. We have incorporated these clarifications into the revised manuscript.
>
> We agree with the reviewer that a comprehensive comparison with recent related work is essential to situate our contributions. To clarify the novelty of our Semantic Representation Attack (SRA), we first categorize prior jailbreaking methods based on their core optimization objective. As shown in our comparison table, existing approaches largely fall into three groups: (1) those that optimize for specific textual patterns (e.g., "Sure, here is..."), (2) those that rely on manually designed prompts, and (3) those that optimize for a score from a judge LLM.
>
> Our approach differs significantly from all three. Unlike methods targeting textual patterns, we directly optimize for the target semantic representation of harmful responses. This avoids constraining the model to a narrow surface form and overcomes the trade-off between prompt naturalness and attack efficacy that limits prior work. Compared to manual methods, our approach is fully automated and more generalizable. Finally, while methods like TAP and PAIR optimize for a judge's score, our SRA targets the underlying semantic space directly, which is a more fundamental and robust objective than a single scalar score. This conceptual shift allows SRA to achieve superior attack efficacy, transferability, naturalness, and efficiency, as demonstrated in our experiments.
>
> |Method|Year|Venue|Optimization Goal|
> |:---|:---|:---|:---|
> |GCG [1]|2023|arXiv|Textual patterns, e.g. "*Sure, here is ...*"|
> |AutoDAN [2]|2024|ICLR|Textual patterns, e.g. "*Sure, here is ...*"|
> |BEAST [3]|2024|ICML|Textual patterns, e.g. "*Sure, here is ...*"|
> |SAA [4] ([1] in W1)|2025|ICLR|Textual patterns, e.g. "*Sure, here is ...*"|
> |AutoDAN-Turbo [5]|2025|ICLR|High ASR strategy|
> |BadRobot [6]|2025|ICLR|Manually design adversarial prompt|
> |Past Tense [7]|2025|ICLR|Manually design past tense adversarial prompt|
> |TAP [8] ([2] in W1)|2024|NeurIPS|Evaluator score|
> |PAIR [9]|2025|SaTML|Score of judge function|
> |RTLM [10]|2022|EMNLP|Generate red teaming test cases with LM|
> |ReG-QA [11] ([3] in W1)|2025|ICLR|Generate adversarial prompt with Q&A AND A&Q loop|
> |Ours|2025|-|Malicious semantic representation|
>
> This clarification has been included in the revised version. We appreciate your understanding regarding the current system limitations for uploading the revision.
>
> We look forward to your response.
>
> Best regards,
>
> The Authors
>
> **References**
>
> [1] Universal and Transferable Adversarial Attacks on Aligned Language Models <https://arxiv.org/pdf/2307.15043>
>
> [2] AutoDAN: Generating Stealthy Jailbreak Prompts on Aligned Large Language Models <https://arxiv.org/pdf/2310.04451>
>
> [3] Fast Adversarial Attacks on Language Models In One GPU Minute <https://arxiv.org/pdf/2402.15570>
>
> [4] Jailbreaking Leading Safety-Aligned LLMs with Simple Adaptive Attacks <https://arxiv.org/abs/2404.02151>
>
> [5] AutoDAN-Turbo: A Lifelong Agent for Strategy Self-Exploration to Jailbreak LLMs <https://arxiv.org/abs/2410.05295>
>
> [6] BadRobot: Jailbreaking LLM-based Embodied AI in the Physical World <https://arxiv.org/abs/2407.20242>
>
> [7] Does Refusal Training in LLMs Generalize to the Past Tense? <https://arxiv.org/abs/2407.11969>
>
> [8] Tree of Attacks: Jailbreaking Black-Box LLMs Automatically <https://arxiv.org/abs/2312.02119>
>
> [9] Jailbreaking Black Box Large Language Models in Twenty Queries <https://arxiv.org/abs/2310.08419>
>
> [10] Red Teaming Language Models with Language Models <https://arxiv.org/pdf/2202.03286>
>
> [11] Does Safety Training of LLMs Generalize to Semantically Related Natural Prompts? <https://arxiv.org/abs/2412.03235>

---

> > ### Comment · Reviewer_oJrc · 2025-08-06
> >
> > Thank you for your response. All my concerns are resolved now and I have updated the score accordingly.

---

> > > ### Author Response · Authors · 2025-08-06
> > >
> > > Thank you for your feedback and for updating the score. We appreciate your time and consideration.

---

### Note · Authors · 2025-08-11

We sincerely thank the reviewers, (senior) area chairs, and program chairs for their thoughtful evaluations and constructive discussions. We are pleased that our methodological rigor, theoretical contributions, and strong empirical results were recognized, and that our additional experiments and clarifications have addressed reviewers' concerns.

###  Key Additions During Rebuttal

- **Clarified novelty**
  Distinguished our *Semantic Representation Attack* (SRA) from prior work by directly optimizing malicious semantic representations — rather than textual patterns or judge scores — supported by a new related‑work comparison table.

- **Expanded robustness evaluation**
  Added results against advanced defenses, with SRA maintaining or improving attack success rates relative to strong baselines.

- **Demonstrated black‑box transferability**
  Showed that SRA‑generated prompts from proxy models achieve **state‑of‑the‑art** success on closed‑source models (e.g., GPT‑3.5/4 Turbo) without requiring logits.

- **Addressed metric concerns**
  Supplemented perplexity‑based coherence evaluation with a **human study** (>95% naturalness ratings) and justified perplexity’s role as a reasonable proxy in this context.

- **Strengthened theory–practice links**
  Made explicit how our theoretical results (e.g., *semantic convergence theorem*) inform parameter choices like the perplexity threshold τ, and provided a practical tuning heuristic.

- **Reported compact prompt lengths**
  Most successful prompts were under **3 tokens**, reinforcing efficiency and stealth.

###  Conclusion

We believe these additions strengthen our contributions:

- Introduced a principled, semantically grounded jailbreak paradigm
- Proved its theoretical basis
- Empirically established its superior **effectiveness, naturalness, efficiency, and robustness** across 18 open‑source and multiple closed‑source LLMs

We thank the committee for their time and consideration, and look forward to contributing this work to the **NeurIPS community**.

---

### Decision · Program_Chairs · 2025-09-17

**Decision:**

Accept (poster)

**Comment:**

This paper proposes a new semantic representation attack for LLMs. The ideas and insights presented are highly impactful, and all reviewers unanimously recommend acceptance.